# Sustainable Approach in IT Project Management—Methodology Choice vs. Client Satisfaction

**Monika Woźniak** 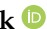

Department of Business Informatics, Faculty of Management, University of Gdańsk, 80-309 Gdańsk, Poland; monika.wozniak@ug.edu.pl

**Abstract:** Major elaborations on sustainable development relating to the general economic sphere of various issue-related areas. Surprisingly little explicit guidance presenting this topic from the organizational side exists. To fill this gap, this paper attempts to propose and verify the concept of a sustainable approach to IT project management by involving the client at the stage of choosing the project management methodology. The research scope constitute IT projects, the subject of which was the development of software commissioned by the organization. The study aims to assess how the internal perspective of sustainability in IT projects, manifested in the matching of IT project management methodology to the client, affects the overall client's satisfaction with the IT project, its products, and results. In the study, 64 IT projects implemented in Polish SME organizations were analyzed throughout their life cycle. The study has proved that introducing and improving the internal perspective of sustainability in IT projects, by matching an IT project management methodology to the client type is one of the key factors determining the level of client satisfaction and thus the assessment of the project's success. The evaluation was made using the Servperv method by developing a measurement instrument dedicated to the IT project management area. Researches can use the proposed approach and the Servperv measuring tool to carry out similar analyses with different sample groups across other countries. The software industry can find results valuable and useful with regard to the IT project management improvement.

**Keywords:** IT project management; sustainable management; success management; software development; alignment IT—business; Servperv

## 1. Introduction

Project success is the prevailing direction of studies in the field of project management [1–6]. There are numerous publications discussing the factors that influence the success and failure of a project [7–10]. These problems are also widely analyzed in the category of IT projects [11–16], with a low rate of project success recorded—similarly to all other project groups [17,18].

Advanced IT project management methodologies and tools, or adaptations of methodology development directions—from different types of classic approaches to many variations of agile methods—have not resulted in any significant increase in success rates for IT projects over recent years [19,20]. The dissatisfaction felt by the end-users of IT projects with regard to their results and the rate of project failures still remain a cause for concern [21,22].

The following key problems have been identified in the field of IT project management:

- Low utilization of IT system functionalities (in most companies, only ca. 10% of the software functionalities are actually used) [23];
- Communication problems with the end users [24];
- Increasing expectations from the business world with regard to flexibility and willingness to change the mindset of the IT sector [25–27];

- It is not technical perfection that is the key skill, but rather the ability to collaborate and build interpersonal relations within the company to enhance its competitive advantage [28–30].

The evolution of the definition of a successful project determines how the factors that influence success or failure are identified. Hence, the "Iron Triangle" criteria (time, cost and scope) have been in force for a long time. In 1999, Atkinson was one of the first to indicate the limitations of this approach [31]. After 2000, the success and failure narrative of the client, together with symbolic and rhetoric success and failure factors recognized by the client have found their way into the criteria used to evaluate a project's success [5,32,33]. These changes are consistent with the CHAOS reports from the Standish Group [34] on the critical success factors of IT projects Clients' expectations are enumerated among major risks of an IT project, while managing them is considered its key success factor. For a few years now, the awareness of the client's role, or rather lack thereof, has been pointed out as number one among both critical success factors and failure causes in IT projects.

The evolution of the definition of a project's success has been empirically confirmed by analyses on the Polish ICT sector. The impact of different factors on the project's success has been evaluated [35]. Taking into consideration the influence of the buyers on project process management is indicated as the strongest determinant of success. 100% of the respondents indicated this as the key factor.

According to Gartner Group and Standish Group reports, user problems have always dominated the problems of IT project management, but over the years the approach to solving these problems has been different. First, there was a strong focus on methodology, tools and technical development. The lack of the expected results meant that mere technical and methodological perfection is becoming less and less important in IT project execution, with soft management mechanisms [36,37] and the recognition of the impact of the users (IT project clients) being of vital importance here [38]. This is a clear indication that without a sustainable approach to these issue-related areas, it is impossible to achieve success in IT projects [39].

IT projects are somewhat idiosyncratic, because of the client's role in the project [40,41]:

- The client is the main source of information—it is the client who specifies the IT project requirements;
- Interaction with the client (method, form, frequency) is a significant contributor to the accuracy and efficiency of defining the client's IT needs;
- The client's role in the evaluation of a project's success—dependent to a large extent on the client's narrative.

Hence, more attention should be devoted to the client's role in IT project management and to selecting an individual approach [42–44].

It directs attention to sustainability in the IT projects area. This issue can be considered from two perspectives—internal and external [45]. The internal perspective refers to processes and areas of IT project management, along the project life cycle. The external perspective refers to the project impact in terms of social and environmental impacts etc. In this article the focus is on the internal perspective of sustainability in IT projects.

The structure of the article is as follows. The next section presents a research background with a literature review. It ends with the definition of the research gap and specifying the research problem in the hypothesis form. In the Section 3, the research method and process are described. The next steps show the main results and their discussion in the context of this research. The article ends with conclusions referring to the importance of a sustainable approach to IT project management. In order to better follow the discourse of the paper, Figure 1 presents a graphic abstract showing the logic of the content of the article.

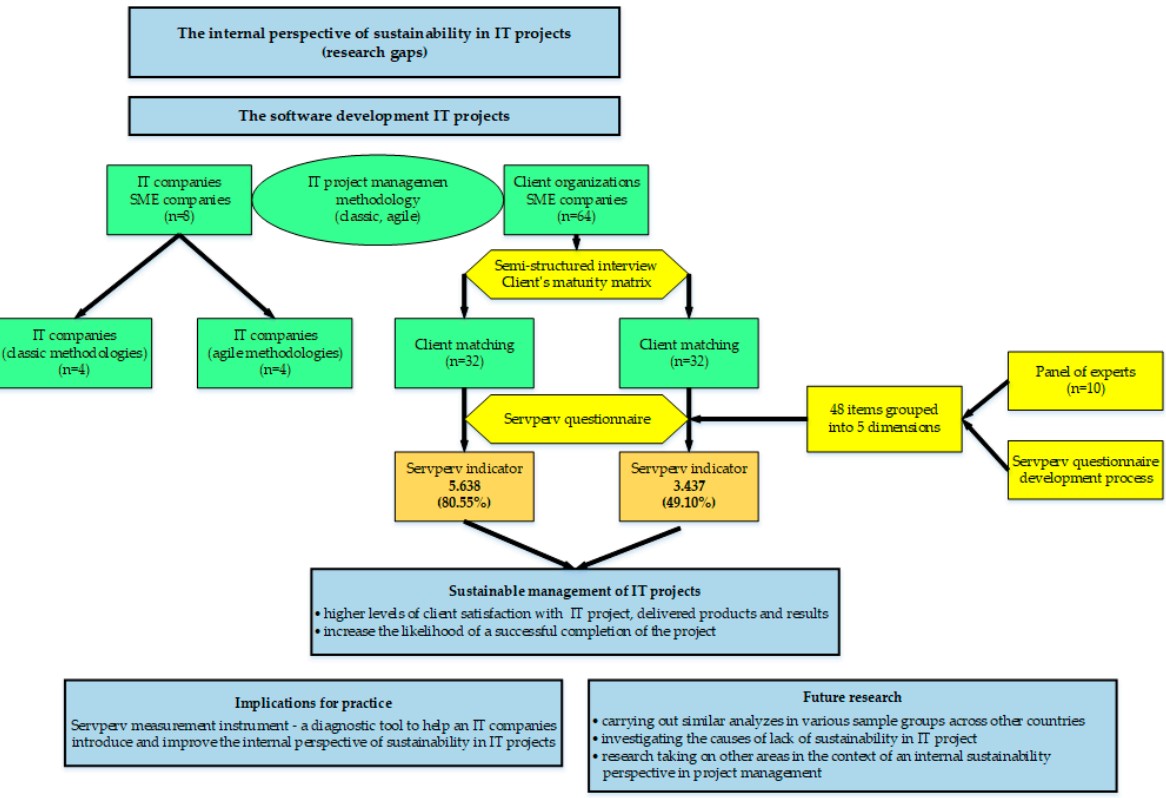

**Figure 1.** The graphic abstract.

## 2. The Research Background and Literature Review

The role of the client in an IT project is a key IT project management problem in the context of the aspects governing a project's success and sustainable approach that are under discussion here. Yet, they cannot be formulated in a mechanical manner, based on methodology standards alone. On the other hand, IT project management practice is still overly oriented toward a procedural and technical approach [46]. As a result, the clients are consulted in an over simplified manner in most projects. The lack of a sustainable approach results in the methodologies chosen for IT projects which usually contradict a client type [47].

In the IT sector, the assessment of client type should be an indispensable element in the process of selecting a suitable IT project implementation methodology—one that will prove effective for co-operation with the particular client in question. No effective management of client relations and expectations is possible without an awareness of categories of clients and without selecting the methodology adequately. This is where the need to diagnose the client type (client maturity) comes in. This kind of diagnosis enables one to identify various client-related factors (client maturity aspects) that are critical to the methodology being considered. Consequently, it answers the question regarding whether the client demonstrates a sufficient level of maturity to meet the requirements of the particular methodology.

Unfortunately, these problems have been neglected in the subject literature, as well as in business practice, where the main focus has been fixed on the methodologies per se and on tools related to IT project [48,49]. Even the line of research on soft factors of IT project management does not consider the client to be a methodology-relevant factor [50,51]. The methodology selection is determined by the project team. This means that IT projects are implemented using the group of methodologies that the team is most used to or most fluent and experienced in [52]. When a methodology selected in this way is suitable for the client type, the final effect is close to the desired one. Most often, however, there are large

incompatibility in this respect. As a result of such a gross lack of sustainable approach to the methodology selection factors many projects are prematurely closed or incompletely delivered.

The topic of sustainability in the area of management is becoming more and more popular. It is discussed from economic, social, environmental, and political perspectives of sustainability [53,54]. However, in the project management field it is a challenge and is considered from the perspective of projects implemented in the above areas [53–55]. In the field of IT projects, the focus is on how to develop projects supporting the areas of sustainable development [56–60]. Even for several years, a special category of IT projects has been developed, the so-called Green IT [61–63]. While there is a lack of studies on how to manage and implement a project in a sustainable way. If anything, sustainability in project management area encompasses the process and knowledge areas of mostly green procurement and partnership (GPP) and project social responsibility (SR) [64–66].

Norms and standards of project management fail to address sustainability or poorly considered environmental sustainability [64]. They often reduce them to a concept of balanced design with the following variables: deviation of the costs, duration, and project quality from the plan specifications [67]. Integrating sustainability and project management poses many difficult issues that, if resolved, can significantly expand the areas to be managed in projects.

Narrowing the topic to the sustainable approach in IT projects, there are only a few studies related to this direction. One of the first studies concerns a sustainability approach to define requirements in ERP projects, where in the early stages of the project, an IT provider and a client organization jointly select process modeling technology, methods, and techniques [68]. This is a case study showing how a technological approach can be used to illustrate the degree of goal mismatch between client and IT providers. This paper is the starting point for developing more sustainable strategies of IT and business cooperation in IT projects.

There are studies addressing the sustainable approach in IT projects as the need to manage an IT project in context. The study reports indicate the context: sociological, government, culture, and politics of the country and a practical consideration to sustainable management of IT projects [69,70] as well as the industry context and its practical consideration to the sustainable implementation of IT project [71]. IT projects are often related to outsourcing issues. Fusiripong, Baharom, and Yusof recognized that in order to ensure sustainable development of IT projects, it is important to select the most suitable IT supplier and proposed standardization of supplier selection criteria for IT outsourcing projects [72]. In their research work, Doherty and Terry combine a sustainable approach in IT projects with the source of their initiation [73]. Their finding indicate that the competitive position of the organization are likely to be more significant and sustainable when the IT project initiative is dictated by the use of existing, complementary organizational resources, rather than resulting directly from the functionality of the IT asset. Paletta and Vieira Junior draw attention to the role of technology itself in sustainable IT project management [74].

There are studies on the field of a sustainable approach, clearly showing the issues of marginalization of the role and involvement of the client in IT projects and its effects in the form of failure of projects because of an inability to capture user insights or respond to user needs. However, these are single case studies showing how to collaborate with clients in the development of a sustainable health IT projects and presenting the practical experiences of dealing with the diversity of requirements [75,76].

The above literature review points out research gaps in two areas:

- The shortage of studies on the client as an important factor in choosing the methodology of IT project management;
- The shortage of studies on how to manage and implement a project in a sustainable way—the internal perspective of sustainability in projects.

The article helps to fill this knowledge gaps in IT project management area, in the category of custom software development and in sustainability field—an internal perspective of sustainability in projects.

The combination of the above gaps shows the need to build a bridge between IT project management and sustainability—an internal perspective of sustainability in project management relating to the client. The internal perspective of sustainability in project management is related to the project management processes and areas, along the project life cycle [45]. The choice of methodology is a key aspect that determines the entire course of an IT project and the production of its products.

The previous discussion leads to the following research hypotheses proposed:

A sustainable approach in IT projects manifests itself in matching the IT project management methodology to the client type which will be positively associated with higher levels of client satisfaction with IT project, delivered products, and results.

The goal of the paper is to verify thus formulated a research hypothesis.

Since the research was undertaken on the Polish basis, the IT industry conditions resulting from this context should also be presented. Polish IT companies from the SME group lack the accredited project management standards (e.g., ISO 10006, ISO 21500, ISO 21502). When implementing IT projects, they are based on the so-called good practices adopted in the industry or developed by the so-called "lesson learn". The standards they consider in the projects serve norms submitted by clients for whom IT projects are implemented. These are standards from the customer's industry to be considered in developing a dedicated software.

Cultural and traditional aspects in IT projects are regulated by the given IT company's code of conduct whose provisions and the adopted organizational culture differ in each company. However, there are certain common elements that also result from the education process in Poland. Polish IT managers are characterized by a technical background. This is due to the educational system, which puts an exclusive emphasis on the technical aspects and skills in the IT domain. Until recently, Polish IT companies that support and deliver business solutions have focused on high technical competences. Currently, they increasingly require understanding of the business area of the organization's core activities and soft skills, which also entails changes in the educational system.

In software development IT projects, it is often important not only the context of the organization in which the project is implemented, but also the country in which it is located [77].

In legal terms, Polish IT companies are struggling with complex and rigid regulations regarding contractual cooperation, which is preferred in IT projects. The form of employment in IT projects requires adjustment to the specificity of work and project requirements. The lack of simpler and more flexible regulations slows down the process of establishing cooperation.

On the other hand, Polish SME companies can count on the support of Polish government in the field of digital development. In January 2021, a new act was introduced in the field of the Public Procurement Law [78]. This act takes into account the needs of the state policy to support the development of small and medium-sized enterprises, as well as innovative, modern products and services. Most of such innovative products and services are related to the IT industry.

In addition, Polish companies can apply for IT projects grants that will improve the organization's processes and increase its efficiency. The Polish Agency for Enterprise Development launches an annual call for proposals in the subsidy path "increasing the digitization level in the company" [79]. This is due to the fact that Poland is trying to be an active partner in building the European Digital Single Market (JRC) [80].

Due to the availability of public aid, new directions in the development of the IT sector and the foreign investment inflow, Polish software and IT services market currently ranks second (after Russia) in IT market in Central and Eastern Europe. All the more, every

effort should be made in the field of education and raising awareness about the need to implement sustainable IT project management.

## 3. The Research Method

To verify the proposed hypothesis, the clients' satisfaction with the IT project, its products, and results were examined. The clients were divided into two groups: group A—clients matching the IT project management methodology used, group B—clients not matching the IT project management methodology used.

The research was conducted in Polish companies from the SME sector within the same category of IT projects—developing dedicated software for organizations. In this paper the client is understood to be the person selected for direct contact with the IT team. The selection of these persons is made by the organization for which the IT project is implemented. These were employees who, in the opinion of senior management, knew the areas of the organization that were relevant to the development of dedicated software. The companies in which the study was conducted were asked to indicate these persons.

In projects where the object is to develop dedicated software for organizations, client maturity is particularly significant. The client maturity assessment is based on a combination of two aspects—awareness and commitment—in the context of an IT project [47].

Awareness in this study is understood as a mental activity, whose level can be marked on a continuum. One end of it stands for a complete awareness of a given domain, problem, project, process, etc., and the other—for a poor knowledge or lack of knowledge of a given domain [81].

Commitment is more complex a notion. According to the groundbreaking research by Alan M. Saks, commitment is connected with employees' feelings as well as the organization itself and the work performed [82,83]. This multidimensional context is adopted in this study.

The aspects referred to above are analyzed using a semi-structured interview in the IT projects where the object is to develop dedicated software for organizations. This tool has been widely used in the research of organization and information systems, from narrower topics such as assessing IS project risk factors [84] to the broader ones such as examining business and IT thinking [85].

The semi-structured interview was designed as a map of topics of importance for a deepened client analysis. An interview form presenting the issues used in the study is included in the supplementary material (title: Semi-structured interview). To reduce the impact of suggestion, no typical question-and-multiple-answers survey was used. Certain topics enabled the interviewer to proceed with an extended discussion. The interview carried out in this manner fully reveals the level of the client's awareness and commitment of a given subject area and eliminates the threat of eliciting information that is more of a client's picture of himself than a reflection of the real situation [86].

Another important and rather sensitive part of the study was the necessity to evaluate the client's responses on the spot and direct further parts of the study accordingly. A semiotic analysis was used for this purpose. The clients' responses were analyzed in terms of both content and expression, which enabled a proper encoding of the information elicited, i.e., ascribing a certain number of points to particular answers in two categories—awareness and commitment. The 0 to 10 scale was applied (where 0—means "none" and 10—means "maximum degree").

The outcomes, when encoded, enable the client type to be identified based on the client maturity assessment matrix (Figure 2).

The combination of the two aspects referred to above—awareness and commitment resulted in four categories of maturity: "high awareness", "high maturity", "low maturity", and "high commitment". Thus, examining the level of awareness and commitment makes it possible to identify the type of client.

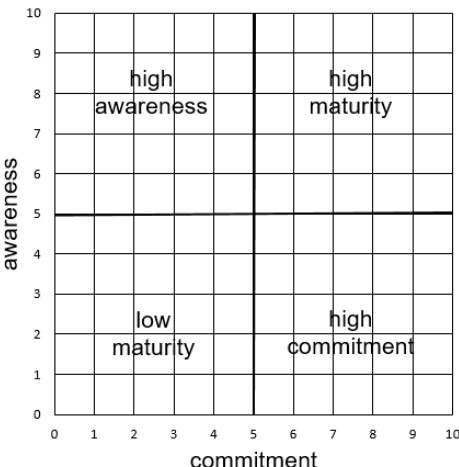

**Figure 2.** The client maturity matrix.

In Polish organizations it has been observed that there are few clients in the "low maturity" category delegated for direct contact with the IT team. In fact, clients belonging to the "high maturity" category are rare in Polish organizations. As a rule, in Polish organizations, employees selected to cooperate in an IT project are clients in the "high awareness" or "high commitment" category. This regularity also emerged in this study. The surveyed clients turned out to be representatives of these two categories. For these categories, the choosing of an adequate methodology is the most important. [87]

The client's maturity juxtaposed with the rules and requirements of IT project management methodologies leads to certain recommendations regarding the matching of IT project management methodologies to the client.

During the research, project management methodologies were divided into two main groups: classic and agile methodologies (light), which are recognized in the subject literature [88]. The group of classical methodologies includes the two most popular standards: PRINCE2 and PMBoK. Both linear and waterfall approaches belong to this group of methods [88]. It should be mentioned here that the development of the traditional current of project management on a global scale is influenced by two main associations: PMI (Project Management Institute) and IPMA (International Project Management Association). They shape both a common international language and standards in project management. PMI and IPMA members and experts from nearly 50 countries contributed to the creation of the ISO 21500 [89] project management standard. This international standard indicates to the model of project integration with the organization's environment, based on the IPMA PMBoK Guide, PRINCE2, and the IPMA competency model.

The agile approach comprises: RUP (Rational Unified Process), Scrum, DSDM (Dynamic Systems Development Method), ASD (Adaptive Software Development), APF (Adaptive Project Framework), as well as Crystal Clear Family, XP (eXtreme Programming). The PMI standard can also be found in this approach. The PMI-ACP spans many approaches to agile such as Scrum, Kanban, Lean, extreme programming (XP) [90].

When assessing the effectives of methodologies (classical, agile) in connection with the client's IT maturity, two quarters are significant: "high awareness" and "high commitment" (Figure 2). This is a result of the specific requirements of these methodologies toward the client, his place, and his role in the IT project [87]. The classical methodology is more effective for the client from the "high awareness" quarter, whereas the agile methodology is more effective for the client from the "high commitment" quarter. In other quarters ("low maturity" and "high maturity") there are no such significant differences between the methodologies.

Thus, on the basis of the semi-structured interview and the client's maturity matrix, it was determined whether the examined client fits the applied IT project management methodology or not. This made it possible to divide clients into the two research groups—A

and B (group A—clients matching the IT project management methodology used, group B—clients not matching the IT project management methodology used).

These instruments can be successfully used to determine which IT project management methodology suite the client and which does not, before choosing the methodology. Undertaking research in this direction at the project initiation stage would enable efficient cooperation with the client and increase the likelihood of a successful completion of the project.

To test clients' satisfaction, a measurement instrument for the IT project management area was developed based on the Servperv method [91]. Servperv is a modification of the Servqual method. Considering the nature of IT projects, where both the process of delivery and the deliverables are equally important, attention was given to the Servqual method [92], and in particular to its performance component, Servperv [91]. This method helps determine how satisfied clients are with services received in a comprehensive manner, to identify all the problems that emerge while providing the service, although it can also be applied to services where tangible products are delivered [93], including IT projects.

The Servqual concept is based on gaps study, and represents the differences between the expectations and perceptions of the service quality actually received. The final model includes the measurement of quality criteria grouped within five dimensions: tangibles, reliability, responsiveness, assurance, and empathy.

The same questionnaire is used to measure both the expected and perceived quality of services. Therefore, the Servqual study is carried out twice. In Servperv, the study takes place only once and concerns only the experienced quality that is the perceived quality [94]. Its authors Cronin and Taylor assumed that researching client's expectations is not necessary because we know that they will always want quality at the highest level. The obtained results can therefore be compared to "ideal quality".

Empirical studies on the use of measuring instruments point to the superiority of Servperv in terms of validity, reliability, and methodological soundness. In addition, it has been proven that Servperv should be the preferred research instrument when comparing service perceptions across different groups [95].

The essence of presented study is the assessment of differences between two groups of clients. Therefore the Servperv method is sufficient for the purpose of this particular study. Consequently, the results were limited to the comparison of the perception of IT projects by both groups. This approach allowed to achieve the research objective—verification of the research hypothesis.

The literature on the subject provides numerous cases of using the Servperv instrument in various fields. The dominant areas are banking [96–100] and tourist and transport services [101–103], but there are also examples of applications in higher education [104].

### 3.1. The Logic of Research—Survey Scheme and Tools

The study was conducted in the period from December 2017 through May 2020 and was divided into the following stages:

- Selecting IT companies using research criteria;
- Analyzing and selecting comparable projects (using below defined criteria) and assessing client type and selection within each of the companies;
- Measuring client satisfaction levels for the IT projects selected.

IT companies were selected as a result of announcing recruitment for research. 48 companies responded to the advertisement, but a detailed analysis allowed to select only 8 companies that met all the assumed criteria important for the study:

- Implementation of IT projects where the object is to develop dedicated software for organizations;
- Using a specific—possible to name IT project management methodology;
- Consent of their clients to conduct research.

Teams working in Polish IT companies from the software industry are usually specialized in one of the IT project management methodologies.

The companies selected for the research sample were divided into two groups according to two principal approaches to IT project management—classic and agile methodologies. Both groups are represented by four IT companies. The vast majority of IT companies in Poland that deal with custom software development are companies from the SME sector. Thus, in this research, all surveyed IT companies belong to the SME sector.

Then, the IT projects implemented by IT companies in client organizations were analyzed. Each IT company's clients were selected so as to represent projects that were similar throughout the study in terms of [52]:

- Size;
- Scope area;
- Complexity;
- IT usage.

The intention behind this approach was to be able to compare the results of analysis.

Most Polish IT projects developing dedicated software for organizations support the area of business processes and are medium size.

In this study complexity is understood related to the level of changes in the organization. Usually, IT projects where the object is to develop dedicated software for organizations impact most elements of the organization and cause a change in the way of working (tactical level of the organization). The level of complexity defined in this way was assumed for IT projects in this study.

As far as the last criterion is concerned—all the analyzed projects were similar in type and involved building an IT system from the ground up. Interaction with the client in the IT project was an important aspect of the study undertaken. It is most clearly visible in IT projects aimed at developing dedicated software for organizations.

The vast majority of Polish companies that prefer dedicated software are companies from the SME sector. All examined clients also represent SME organizations.

As mentioned above, 8 SME companies from the Polish software industry were selected for the analysis: 4 using classic and 4 favoring agile methodologies. Then, the IT projects implemented by them in client organizations were analyzed. Those that met the above assumed criteria were selected. Then, clients from organizations in which selected IT projects have been implemented were examined. Clients of each of the companies were assessed using a semi-structured interview and the client maturity matrix. 8 clients from each company were selected: 4 matching the IT project management methodology used by the company and 4 not matching it. In combination they formed two groups of clients: group A—clients matching the IT project management methodology used, group B—clients not matching the IT project management methodology used. Both groups are represented by 32 clients (of which 16 are serviced by the classic methodology and 16—by the agile methodology). A total of 64 units participated in the Servperv survey (32 units in each group).

Figure 3 presents the study scheme applied in this research work.

### 3.2. The Servperv Questionnaire

The Servperv questionnaire survey was employed to measure the clients' perception toward 5 service quality dimensions and their satisfaction in IT project management area. The Servperv quality dimensions, covering theorems related to clients' perception of IT project management, were developed based on an extensive literature review and research experience of the author in the IT project management area in collaboration with ten experts in IT project management (experienced project managers) and the questionnaire amended accordingly. Care was taken to recruit panelists with sufficient knowledge [105] and within the recommended panel size of 8–12 experts [106,107].

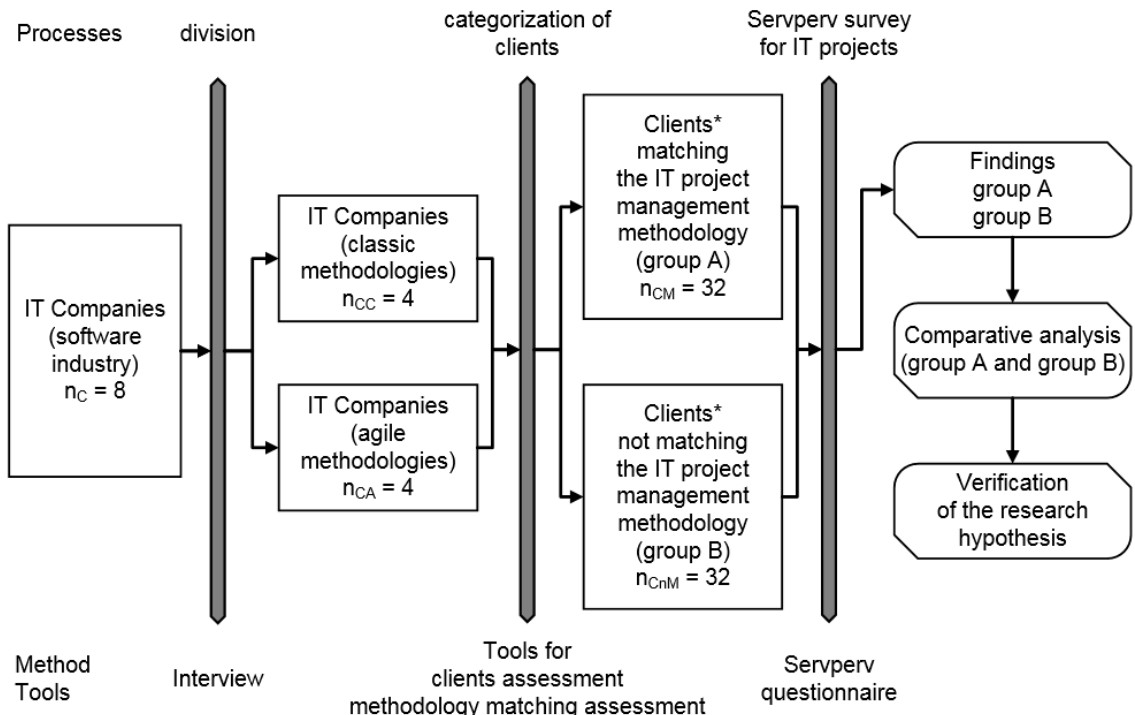

* The companies—recipients of IT projects (building an IT system from the ground up).

**Figure 3.** The study scheme—the choice of IT project management methodology vs. client satisfaction.

Experts were selected as a result of announcing the recruitment for consultation on the development of the measurement instrument dedicated to the IT project management area. Ten IT project managers meeting the following application criteria were selected for this venture:

- Experience in managing IT projects for over 15 years;
- Ongoing management of IT projects, where the object is to develop dedicated software for organizations;
- Experience in applying IT project management methodologies from both groups—classic and agile;
- Ongoing education in project management methodologies confirmed by certificates;
- Direct contact with the client in IT projects.

In addition, panelists had professional or academic connections with research teams during their careers. The criteria were selected so that the knowledge and experience of the panel of experts could provide the best specification of aspects related to the perception of IT project management by clients in the area of developing dedicated software for organizations.

The originally SERVQUAL instrument developed to assess service quality in general include the five following dimensions: tangibles, reliability, responsiveness, assurance, empathy [92]. Each dimension contains adequate theorems. There are 22 of them in total. The literature indicates the possibility of modifying the wording of the instrument to better accommodate its use in an examined subject context [108]. As a result of the discussion with the panel of experts, 5 key areas adequate for the assessment of client satisfaction in software development IT projects were established:

- Tangibles: relates to the evaluation of project documentation, materials, hardware and IT architecture used by the IT company, delivered products;
- Activities: way of performing the services and cooperation with the client;
- Effectiveness: used effective techniques, tools to conduct the project;

- Competencies: aspects proving the appropriate level of knowledge and skills of the IT company for the implementing project;
- Empathy and Individual Approach: understanding and delivering client specific needs.

In the next steps, adequate theorems were selected to allow for the assessment of client satisfaction in the proposed dimensions. This process was performed according to a modified Delphi method. At first, experts were invited to discuss the theorems describing each of the dimensions. Then, the collected theorems were sorted out by eliminating redundancy and combining those relating to the same groups of aspects. A questionnaire was developed with cleaned, structured theorems in the established dimensions. Then two rounds of sessions with expert panels were carried out. A panel of experts assessed the adequacy of the theorems in each of the dimensions. All the respondents were blinded for each other opinions. After the 1st round the results were assessed. Each panelist received an individualized document showing the distribution of all the experts' first round ratings, together with his/her own specific ratings. With this information, in the 2nd round, the panelists were able to respond again to those questions that did not reach consensus. The results of the second round were presented to the panelists and finally approved.

Assessment was made using a nine-point Likert scale, where 1 means fully inadequate and 9 fully adequate. According to the RAND/UCLA appropriateness method [109], the scale was structured in three groups, according to the level of adequacy—inadequacy of the item: from 1 to 3 means "inadequate", from 4 to 6 "uncertain", and from 7 to 9 "adequate". The results of the survey were evaluated through the score and the level of consensus reached. The consensus criteria was established according to a study by Leape, Hilborne and Kahan et al. [110]. The consensus considered as reached, when the 80 percent of panelists indication the 3-point region (1–3, 4–6, 7–9) containing the median. Reaching consensus was considered as "agreement" and a lack of consensus as "disagreement". Items with median scores in the 1–3 range were classified as inadequate, those in the 4–6 range as uncertain, and those in the 7–9 range as adequate. However, all items rated to be "disagreement" for consensus, whatever the median, were classified as uncertain.

In round 1, consensus was reached on 43 of 53 items, all in the range as adequate. Ten items were classified as uncertain and were returned for reconsideration in round 2. The consensus was reached in 5 of those 10 items. Thus, after two rounds, a consensus was reached in 48 items. These 48 items, all in the range as adequate, have been selected to be included in this study.

Collaboration with experts experienced in both method categories resulted in the development of a Servperv measuring instrument that can be used to examine the level of client satisfaction with IT project management. Finally, the Servperv questionnaire consists of 48 theorems grouped into 5 dimensions: tangibles, activities, effectiveness, competencies, empathy, and individual approach (Table 1).

All the clients who qualified for the survey addressed each of the areas by assessing each of the statements in the context of the IT project delivered to them. The statements were rated using a 7-point Likert scale, where 1 represented the respondent's strong disagreement with the statement and 7 indicated a strong agreement [111].

### 3.3. Consistentility and Reliability of the Instrument

The reliability of instrument was tested through internal consistency. The most popular test of internal consistency reliability is the Cronbach's alpha [112]. Cronbach's alpha measures the internal consistency or reliability of a research instrument, that is, it estimates how reliable the answers of the questionnaire or questionnaire domain are. One of the common problems in this area is determining a sufficient sample size. Discussion on Cronbach's alpha test in terms of applications and sufficient sample size so that research conducted can show consistency or stability of an instrument has been widely discussed in the literatures [113–118]. The literature on the subject indicates that the sample size is at least 30 and this condition is met in this analysis.

**Table 1.** Servperv questionnaire for the survey.

| Dimensions | Item No. | Theorem of Servperv Questionnaire |
|---|---|---|
| Tangibles | Tan1 | The project is consistent with the signed contract |
| | Tan2 | The documentation is complete |
| | Tan3 | The materials provided by the IT company are useful |
| | Tan4 | The IT company used adequate hardware and IT architecture |
| | Tan5 | The delivered products are free of errors |
| | Tan6 | The delivered products meet the expectations |
| Activities | Act1 | Activities in the project were consistent with each other |
| | Act2 | The project had adaptation activities (e.g., adaptation of project management processes to the client's company architecture) |
| | Act3 | The project had participative activities—including the client |
| | Act4 | The IT company provided assistance in managing change relation with the project (adaptation policy) |
| | Act5 | The employees of the IT company gave information about the deadline for the implementation of products/services |
| | Act6 | Tasks were carried out in accordance with certain standards |
| | Act7 | Activities related to integration requirements (regarding data, processes, etc.) were carried out |
| | Act8 | There were opportunities for discussion |
| | Act9 | There was a functioning system for communication between stakeholders |
| Effectiveness | Eff1 | What the IT company promised and what they actually delivered was consistent |
| | Eff2 | The IT company ensured the accuracy and transparency of information provided within the scope of the project |
| | Eff3 | There was no scope creep in the project |
| | Eff4 | The IT company identified all project stakeholders while performing a needs analysis |
| | Eff5 | The IT company used effective techniques to conduct the needs analysis (e.g., interview, workshop, observation, etc.) |
| | Eff6 | The IT company delivered services on time |
| | Eff7 | The project results match the required functionalities |
| | Eff8 | There were no disruptions during the project |
| | Eff9 | The method of managing the project gave a sense of stability and continuity in the day-to-day running of the client's company |
| | Eff10 | The IT company created a final product that will allow consistency in business operations and will not disrupt the business operations after its implementation |
| | Eff11 | The IT company was truly committed to the project |
| | Eff12 | The IT team quickly responded to expectations |
| Competencies | Com1 | The IT company clarified the goals of the IT project |
| | Com2 | The IT company aligned/helped to align the IT requirements to the business goals |
| | Com3 | The IT company made high-level rational analyses |
| | Com4 | The IT company also included non-functional (qualitative) requirements in the project |
| | Com5 | The IT company provided clear guidelines that led to production results |
| | Com6 | The IT company used creative problem solving and innovative solutions |
| | Com7 | IT company has secured continuity of adaptation |
| | Com8 | The IT company met the expectations as to the course of cooperation—the cooperation model |
| | Com9 | The IT company provided mentoring services for the client |
| | Com10 | The IT team displayed satisfactory knowledge and skills |
| | Com11 | IT company employees had a sense of responsibility |

**Table 1.** *Cont.*

| Dimensions | Item No. | Theorem of Servperv Questionnaire |
|---|---|---|
| Empathy and Individual Approach | EmInAp1 | The behavior of the IT team inspired confidence in the client |
| | EmInAp2 | The IT company made the client feel secure |
| | EmInAp3 | The IT company treats the client individually |
| | EmInAp4 | IT company employees understand the specific needs of the client |
| | EmInAp5 | Work in the project was carried out in a way that was convenient for the client |
| | EmInAp6 | The IT team expressed themselves comprehensibly to the client |
| | EmInAp7 | The IT company sought to resolve conflict and build consensus |
| | EmInAp8 | Some members of the IT team paid the client special attention during the project (client guardians) |
| | EmInAp9 | Some members of the IT team were always willing to help the client |
| | EmInAp10 | There was a visible sense of commitment from the IT company towards the project |

The reliability analysis results are summarized in Table 2. The Cronbach's alpha value for all constructs ranges between 0.756 and 0.994. All the values are above the value of 0.70 thus demonstrate that the scales are consistent and reliable.

**Table 2.** Reliability analysis result.

| Dimensions | Cronbach's Alpha Group A | Cronbach's Alpha Group B | N of Items |
|---|---|---|---|
| Tangibles | 0.756 | 0.901 | 6 |
| Activities | 0.961 | 0.936 | 9 |
| Effectiveness | 0.984 | 0.974 | 12 |
| Competencies | 0.975 | 0.989 | 11 |
| Empathy and individual approach | 0.994 | 0.992 | 10 |

Data was analyzed using the Statistica software version 13.3. The descriptive statistics provided data summary in terms of mean value and standard deviation. In order to identify the significance of differences between two groups of respondents in the adopted dimensions, Z-test two means was performed. In addition, the Servperv indicator was computed for both groups.

## 4. Research Results

The data collected from Servperv questionnaires together with some discussion are reported in this section.

Figure 4 presents the mean score of 48 IT project quality items for both groups. For clients whose profile was compatible with the methodology used (group A) the mean score ranges from 4.78 the lowest to 6.19 the highest. For clients whose profile was incompatible with the methodology used (group B) the lowest mean score is 2.94 and the highest one, 4.16. Detailed research results are included in the supplementary material (title: Study-results).

The above 48 IT project quality items were grouped into 5 Servperv dimensions: tangibles, activities, effectiveness, competencies, empathy, and individual approach. A comparative analysis of two groups A and B was made in terms of these dimensions, according to the Servperv method (Table 3).

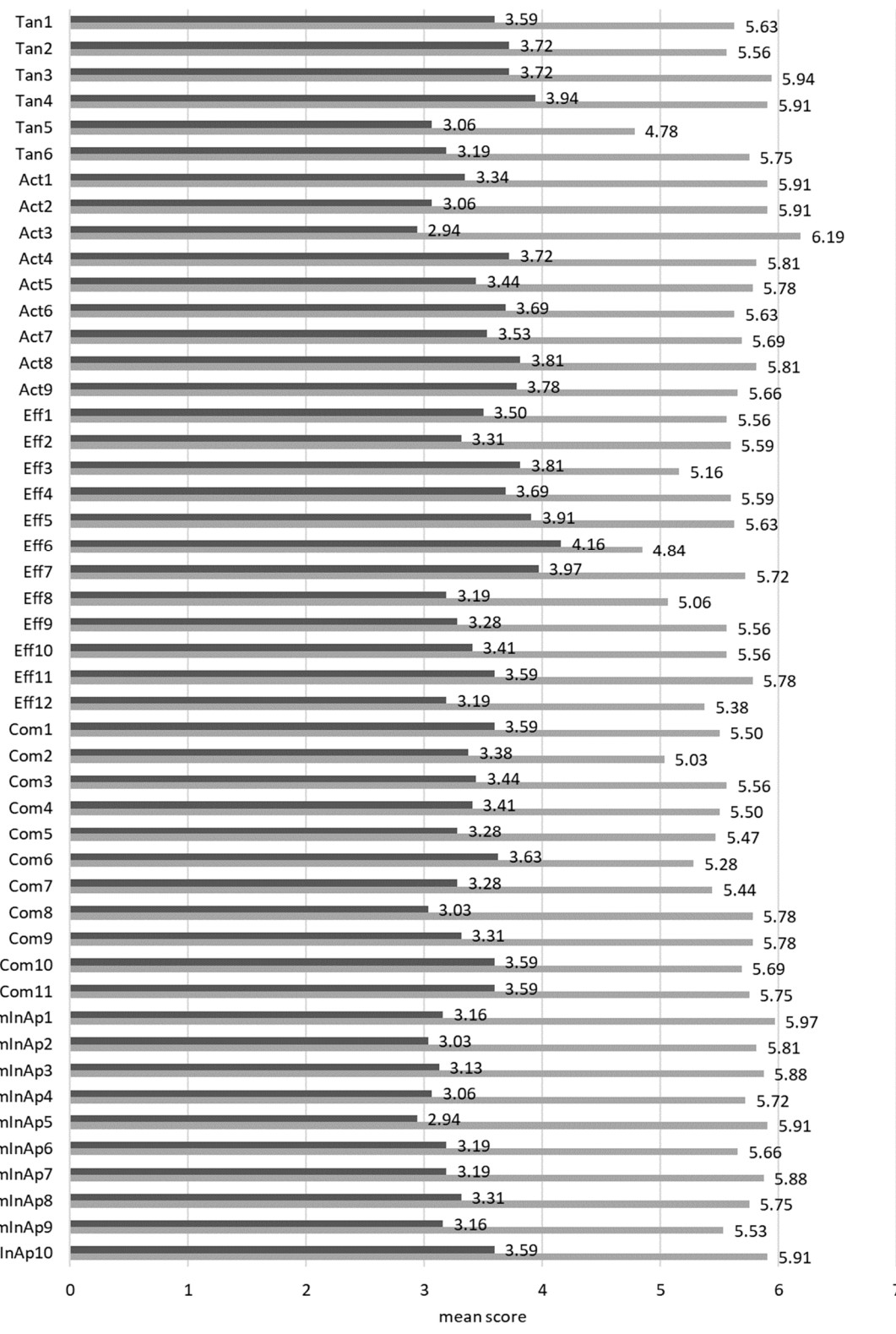

**Figure 4.** Mean score of the Servperv survey.

**Table 3.** Comparison of the dimensions and Servperv indicator ratings for two groups of clients.

| Dimension | Group A | | Group B | | Z-Test |
| --- | --- | --- | --- | --- | --- |
| | Mean Score | SD | Mean Score | SD | |
| Tangibles | 5.594 | 0.425 | 3.536 | 0.340 | 9.265 * |
| Activities | 5.819 | 0.171 | 3.479 | 0.315 | 19.596 * |
| Effectiveness | 5.453 | 0.286 | 3.583 | 0.325 | 14.960 * |
| Competencies | 5.526 | 0.229 | 3.412 | 0.183 | 23.896 * |
| Empathy and individual approach | 5.800 | 0.135 | 3.175 | 0.179 | 37.043 * |
| Servperv indicator | 5.638 | | 3.437 | | |

Notes: Significant at * <0.05.

For all areas the dimension, the difference between mean score for group A and B, is significant (see Z-test Table 3). This means that client satisfaction with the IT project in both groups differs.

Obtained final Servperv indicators for the two groups indicate a significant differences between them. It can be converted into the following percentages:

- ServpervA = 80.55% for the client matching the IT project management methodology used,
- ServpervB = 49.10% for the client not matching the IT project management methodology used.

The Servperv indicator shows a difference of over 30% between the groups, in favor of clients compatible with the IT project management methodology. The evaluation of IT projects managed by matching the methodology with the type of client ranks quite highly—above 80%. However, in the group of clients in which the methodology was not compliant with the client type, this evaluation was only 49%.

The biggest differences between the two groups occurred in the dimensions of empathy and responsiveness. In the group of clients not matching the applied methodology in the IT project, the dimensions of empathy and responsiveness was rated the lowest. The same dimensions, in turn, obtained the highest quotations in the group of clients matching the methodology used.

The projects implemented using a methodology compatible with the client type manifest much higher client satisfaction levels. These results clearly support the research hypothesis formulated at the beginning of the paper. The results of this study interpreted in the context of the contemporary understanding of project success [33] and sustainable management [119] show that matching an IT project management methodology to the client type significantly enhances the probability of a successful project.

## 5. Discussion and Conclusions

The choice of methodology is a key aspect that determines the course of an IT project and the production of its products, and is usually made by the IT team without the client. This article attempts to verify whether a sustainable approach in IT projects manifests itself in the IT project management methodology with regard to the client type, affects the overall client's satisfaction with IT project, delivered products and results. The survey was conducted in Polish companies from the SME sector. The study compares client satisfaction indicators across two groups—clients matching the IT project management methodology used and clients not matching the IT project management methodology used. The presented research results verified that the research hypothesis was significant and allowed to achieve a research goal.

This article makes a new contribution to the current literature review by proposing and verifying the approach to a sustainable approach in IT projects. It completes the

research gap in two knowledge domains: the IT project management and sustainability from the internal project sustainability perspective.

The study has proved that matching an IT project management methodology to the client type is one of the key factors that influences a project's success by the clients. Without applying a sustainable approach, it is impossible to achieve complete success in IT projects. Sustainable approach in the form of the matching the IT project management methodology to the type of client is consistent with the developing, new area of the PM knowledge—Success Management. Seeking better sustainable client co-operation in IT projects leads to the enhanced integration of organizational growth. In effect, improvement will be achieved not only in the field of information and technology, all well as in aspects related to organization, management, and culture. By following this direction, it will be possible to achieve flexibility and adapt better to changes occurring as a result of the implementation of a given IT project, while also enhancing the overall performance of the business unit.

### 5.1. Implications for Practice

The research demonstrates that the lack of sustainability in IT project management is also reflected in the practical stage. This subject is also underrepresented in the IT project management standards, hence the urgency for publications to promote a new sustainable approach to the client in IT projects.

Using the existing path of methodological and technological evolution as a basis for IT projects is not sufficient any longer. The client's role in the project has become a critical element of IT projects. This element has to be addressed in a new way. Therefore, most IT companies face a major challenge applying an internal sustainability perspective to their IT projects.

In most organizations, IT projects trigger non-concurrent changes, thereby reducing the effectiveness of the project in general and generating tension. By introducing a sustainable approach in IT projects through assessment of client type and select a suitable project management methodology, better communication, collaboration, and integration is achieved among those responsible for the business and IT specialists, as well as in the field of software development. In this way, changes related to the IT project may be efficiently integrated with the organization's growth, instead of bringing about a decline in its performance [120].

The present study verifies the suggested sustainable approach to the selection of an IT project management methodology that is suited to the client type. This assessment is applied as a measuring instrument for the IT projects domain, created for the purpose of this study, grounds in the Servpera method which can be used as a diagnostic tool by IT companies to help introduce and improve the internal perspective of sustainability in IT projects.

### 5.2. Limitations and Future Research

The study is limited to Polish SME companies. It would be worth carrying out similar analyses in various sample groups across other countries. Such a research would allow for conducting comparative studies that would demonstrate differences and similarities between countries, shading light on the issues raised.

The article does not consider other perspectives of sustainable IT project management in the context of client interaction. In further studies, one could refer to the perspective of this imbalance causality, e.g., the competency gaps of IT project managers in the field of five key competences [63], namely: system thinking competences, anticipatory competences, motivational competences, strategic, and interpersonal competences. Looking for causality in this area can constitute an relevant research problem.

In addition, this article focuses on the neuralgic aspect of IT projects with regard to client tailored software development. Further elaborations could take on other areas in the context of an internal sustainability perspective in project management. First, it seems interesting to discuss the study results assessing whether and to what extent IT projects

are managed in a sustainable manner across different countries. Second, relationship between project sustainability management and project success can become a focus of further studies. Such research would fill an important gap in the literature regarding the internal sustainability perspective of projects, and, at the same time, provide a research framework for projects in other areas.

**Supplementary Materials:** The following are available online at https://www.mdpi.com/2071-105 0/13/3/1466/s1, Semi-structured interview; Study-results.

**Funding:** This research received no external funding.

**Institutional Review Board Statement:** Not applicable.

**Informed Consent Statement:** Not applicable.

**Data Availability Statement:** Not applicable.

**Conflicts of Interest:** The author declare no conflict of interest.

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
