# Peer review of "Sustainable Approach in IT Project Management—Methodology Choice vs. Client Satisfaction"

_sustainability, doi:10.3390/su13031466_

Round 1

Reviewer 1 Report

A more extensive discussion of the research methodology would add value to the article

Author Response

Response to Reviewer 1 Comments

Thank you very much for your effort in reviewing my manuscript.

Thank you both for your positive assessment of my paper and for suggestion to improve the research methods section.

I have addressed your comment - suggestion, as described below.

Point 1: A more extensive discussion of the research methodology would add value to the article

Response 1: The research methods section has been significantly enriched with an explanation of the method of client maturity assessment. The description of the instruments used the study (semi-structured interview and client maturity matrix) was made more detailed. The structure of the client maturity matrix is described (Lines: 191-199, 221-231) and presented in a new figure (Lines: 218-221). The using of a semi-structured interview is described (Lines: 204-216), and the interview itself is included in the supplementary materials (Line: 206-207). It was explained how the client maturity matrix can be related to methods (classic, agile), referring to previous research (Lines: 244-250). Finally, it was summarized how the clients were divided into two groups, A and B (Lines: 251-255).

Then, the develop of a research instrument based on the Servperv method was justified (Lines: 278-288). The Servperv method is successfully used in contemporary research as a basis for constructing own measuring instruments in various areas. Examples of contemporary Servperv applications in various areas have been added in the manuscript (Lines: 286-288).

Then, the process of selecting companies for the study is presented in more detail, their characteristics and the selection criteria used are described. The criteria and method of selecting IT companies for the study are described in the manuscript in lines: 296-302. The criteria and method of selecting clients for the study were determined by the criteria for selecting IT projects, described in lines: 303-329. Finally, the research process and number of the units studied were explained in more detail (Lines: 330-340).

The research problem was formulated in the summary of the research background. In the research background, a literature review has been added, which shows how the topic of sustainability in projects is discussed. In this way, the research gaps were identified and then the research problem was specified in the context of the identified gaps. (Lines: 120-174)

The verification of the research hypothesis was based on comparing the differences between two independent groups. To illustrate the differences between them, a comparison of averages in the  dimensions of Servperv instrument was used. A detailed study results were not included with the manu.script. They have been included as supplementary material. (Lines: 400-401)

The reason and justification for the use of Cronbach's alpha is more explained (Lines: 377-384).

A Conclusion section has been added to the Discussion section (Lines: 433-457) and new subsections are introduced there: Implications for practice (Lines: 458-477); Limitations and future research (Lines: 478-496).

All changes to the manuscript are marked in green.

Reviewer 2 Report

The research work claims to propose a sustainable approach in IT project management method choice vs. client satisfaction.

The abstract starts by presenting the perspective of "micro". However, in the paper, such a framework cannot be found. 

Furthermore, numerous key concepts are not defined: what does "sustainable" mean in this context? what does "complexity" mean in this context? 

The results are based on interview. The interview questions are provided, but the answers are not provided... this makes it impossible to verify the results presented. This invalidates the research. 

Author Response

Response to Reviewer 2 Comments

Thank you very much for your effort in reviewing my manuscript. Thank you for pointing out weaknesses and suggestions for improving them.

I have read all of your comments and claims very carefully and have tried to resolve all reported problems and doubts.

I have addressed all the issues you raised, as described below.

Point 1: The abstract starts by presenting the perspective of "micro". However, in the paper, such a framework cannot be found.

Response 1: Indeed, the use of the terms "macro" and "micro" in this context may cause confusion. This have been changed in the manuscript. Reference was made to the general economic sphere and organizational perspective - sustainable management, which better capture the issues of paper (Lines: 8-12).

Point 2: Furthermore, numerous key concepts are not defined: what does "sustainable" mean in this context? what does "complexity" mean in this context? 

Response 2: I admit it has been neglected. The key concepts raised have been explained in the manuscript.

“Sustainable”  in the context of IT project management means orientation not only on the ‘inner’ benefits of the project, but also to the benefits that it should produces towards the clients that will interact with its deliverables. It will be possible when the IT project management methodology will also match the client, not just the project team. (Lines: 83-87, 168-173)

"Complexity" is used to mean the level of complexity of an IT project, related to the level of changes in the organization. The study relates to projects where the object is to design dedicated software for organizations. Projects that impact most elements of the organization and cause a change in the way of working were qualified for the study. (Lines: 320-323)

“Client” - in this paper the client is understood to be the person selected for direct contact with the IT team. These were employees who, in the opinion of senior management, knew the areas of the organization that were relevant to the development of dedicated software. (Lines: 185-190)

“Client maturity”  is a combination of two aspects—awareness and commitment—in the context of an IT project, which were presented in the manuscript in the lines: 191-199, 221-224.

“Expert” - an expert is understood as an experienced IT project manager who meets the assumed criteria, which were presented in the manuscript in the lines: 353-366.

Point 3: The results are based on interview. The interview questions are provided, but the answers are not provided... this makes it impossible to verify the results presented. This invalidates the research. 

Response 3: Thank you very much for drawing attention to this important shortcoming. Indeed, detailed study results were not included with the manuscript. They have been included as supplementary material (Lines: 400-401).

Some sections of the paper have been significantly changed.

In Abstract and Introduction the purpose, subject, results and conclusions have been clearly define in the context of sustainability (Lines: 8-24, 83-87). In the research background section, a new literature review has been added (Lines: 120-159). This literature review shows how the topic of sustainability in projects is discussed, with particular attention to the internal perspective of sustainability in IT projects. It allowed to present a current research gaps in this area (Lines: 160-173). The research problem was specified in the context of the identified gaps (Lines: 174-178).

The research methods section has been significantly enriched with an explanation of the method of client maturity assessment. The description of the instruments used the study (semi-structured interview and client maturity matrix) was made more detailed. The structure of the client maturity matrix is described (Lines: 191-199, 221-231) and presented in a new figure (Lines: 218-221). The using of a semi-structured interview is described (Lines: 204-216), and the interview itself is included in the supplementary materials (Line: 206-207). It was explained how the client maturity matrix can be related to methods (classic, agile), referring to previous research (Lines: 244-250). Finally, it was summarized how the clients were divided into two groups, A and B (Lines: 251-255).

Then, the develop of a research instrument based on the Servperv method was justified (Lines: 278-288). The Servperv method is successfully used in contemporary research as a basis for constructing own measuring instruments in various areas. Examples of contemporary Servperv applications in various areas have been added in the manuscript (Lines: 286-288).

Then, the process of selecting companies for the study is presented in more detail, their characteristics and the selection criteria used are described. The criteria and method of selecting IT companies for the study are described in the manuscript in lines: 296-302. The criteria and method of selecting clients for the study were determined by the criteria for selecting IT projects, described in lines: 303-329. Finally, the research process and number of the units studied were explained in more detail (Lines: 330-340).

The verification of the research hypothesis was based on comparing the differences between two independent groups. To illustrate the differences between them, a comparison of averages in the  dimensions of Servperv instrument was used. A detailed study results were not included with the manuscript. They have been included as supplementary material (Lines: 400-401).

The reason and justification for the use of Cronbach's alpha is more explained (Lines: 377-384).

A Conclusion section has been added to the Discussion section (Lines: 433-457) and new subsections are introduced there: Implications for practice (Lines: 458-477); Limitations and future research (Lines: 478-496).

All changes to the manuscript are marked in green.

Reviewer 3 Report

Title:

OK. The topic shown in the title is very interesting. This suggests to keep reading.

Abstract:

Macro and micro should be better explained. The micro perspective is not understood.

Servperv should be better introduced.

Keywords:

"Agile" is not mentioned in the title nor the abstract.

Introduction:

line 37...only ca.37 ?

Polish context should be better introduced. Only reference [34] has been used to explain it. In addition, Polish environment has not been included into the abstract.

Please, if you use the term "client", who is the customer? Try not to confuse and/or mix them.

At the end of the introduction, the publisher recommends to include the content of the next sections.

The research method:

line 111...which clients?

Steps to be done must be previously explained.

Clients matching the IT project management methodology used and clients NOT matching the IT project management methodology used should be better explained and contextualized.

Servperv method should be justified, explained and referred from line 120 to 114 (first time). 

Criteria to assess client maturity (based on increasing awareness and commitment) should be explained for applying in this case. This is not the introduction but the research method, so this should be solved. Are you dividing awareness and maturity only into low and high? Which are the requirements to be included into one or another?

With the term "classic", do you refer to waterfall approaches? Please, try to specify it. Waterfall is stills valid and very very used in other sectors. Among agile approaches you forget the PMI one.

4 companies with classic methods and 4 with agile ones. 32 clients matching and 32 not matching....but which ones come from classic or agile? 16 per each?

More applications of Servperv should be included in order to consider this method.

These A and B groups and divided according to their match with IT PM methodologies but, other characteristics from their companies (size, level of technology applied, investments, etc.) should be stressed in order to chech if the two sample of 32 clients are ok.

Research results:

A simple average study...

Discussion:

Sustainable approach is absolutely missing. 

Author Response

Response to Reviewer 3 Comments

Thank you very much for your effort in reviewing my manuscript. Thank you both for your positive words and for pointing out weaknesses and suggestions for improving them.

I have read all of your comments and claims very carefully and have tried to resolve all reported problems and doubts.

I have addressed all the issues you raised, as described below.
All changes to the manuscript are marked in green.

Point 1:Abstract:

Macro and micro should be better explained. The micro perspective is not understood.

Response 1: Indeed, the use of the terms "macro" and "micro" in this context may cause confusion. This have been changed in the manuscript. Reference was made to the general economic sphere and organizational perspective - sustainable management, which better capture the issues of paper (Lines: 8-11).

Point 2: Abstract:

Servperv should be better introduced.

Response 2: Agree, it was not very clear Serperv introduction.

A better explanation of the use of the Servperv method in the abstract has been included (Line: 24-27).

Point 3: Keywords:

"Agile" is not mentioned in the title nor the abstract.

Response 3: Indeed, it should be opt out of that keyword. The word "Agile" has been deleted.

Point 4: Introduction:

line 37...only ca.37 ? Polish context should be better introduced. Only reference [34] has been used to explain it.

Response 4: Indeed, the Polish context should be shown more. The Polish context was better described in the research method, in the description of the client's maturity in Polish organizations (Lines: 225-228). The Polish context have been also included in both the characteristics of IT companies (Lines: 303-309) and the characteristics of client organizations (Lines: 318-319, 328-329).

Point 5: Introduction:

In addition, Polish environment has not been included into the abstract.

Response 5: Indeed, information about the Polish environment has been omitted in the abstract. This information has been added (Lines: 20-21).

Point 6: Introduction:

Please, if you use the term "client", who is the customer? Try not to confuse and/or mix them.

Response 6: The terms "client" and "customer" are often used interchangeably in literature, especially in the IT area. Therefore, one term was chosen "client" and its use was better defined in the context of the presented research (184-190). The term "customer" has been changed throughout the text to "client".

Point 7: Introduction:

At the end of the introduction, the publisher recommends to include the content of the next sections.

Response 7: The structure of the article at the end of the Introduction has been described (Lines: 88-92).

Point 8: The research method:

line 111...which clients?

Steps to be done must be previously explained.

Clients matching the IT project management methodology used and clients NOT matching the IT project management methodology used should be better explained and contextualized.

Response 8: This recommendation was applied and the research method section have been redrafted. At the beginning it was more explained who the clients are in the study (Lines: 194-199, 204-234, 244-250), and then how they were divided into two compared categories.

On the basis of the semi-structured interview and the client's maturity matrix, it was determined whether the examined client fits the applied IT project management methodology or not. This made it possible to divide clients into the two research groups - A and B (group A—clients matching the IT project management methodology used, group B—clients not matching the IT project management methodology used) (Lines: 251-255).

The instruments used in the study: semi-structured interview and client maturity matrix were presented. The structure of the client maturity matrix is described (Lines: 191-199, 221-231) and presented in a new figure (Lines: 218-221). The using of a semi-structured interview is described (Lines: 204-216), and the interview itself is included in the supplementary materials (Line: 206-207). It was explained how the client maturity matrix can be related to methods (classic, agile), referring to previous research (Lines: 244-250). Finally, it was summarized how the clients were divided into two groups, A and B (Lines: 251-255).

Point 9: The research method:

Servperv method should be justified, explained and referred from line 120 to 114 (first time). 

Response 9: In the research method section, suggested modifications were made to the description of the Servperv method. At first time of using the word Servperv literature source has been added (Line: 261).

Point 10: The research method:

Criteria to assess client maturity (based on increasing awareness and commitment) should be explained for applying in this case. This is not the introduction but the research method, so this should be solved. Are you dividing awareness and maturity only into low and high? Which are the requirements to be included into one or another?

Response 10: The research methods section has been significantly enriched with an explanation of the method of client maturity assessment. The description of the instruments used the study (semi-structured interview and client maturity matrix) was made more detailed. The structure of the client maturity matrix is described (Lines: 191-199, 221-231) and presented in a new figure (Lines: 218-221). The using of a semi-structured interview is described (Lines: 204-216), and the interview itself is included in the supplementary materials (Line: 206-207). It was explained how the client maturity matrix can be related to methods (classic, agile), referring to previous research (Lines: 244-250). Finally, it was summarized how the clients were divided into two groups, A and B (Lines: 251-255).

Point 11: The research method:

With the term "classic", do you refer to waterfall approaches? Please, try to specify it. Waterfall is stills valid and very very used in other sectors. Among agile approaches you forget the PMI one.

Response 11: Indeed, it was not included in the description of the classical methods. Approaches in classical methods have been specified (Lines: 237-238).

Thank you for reminding PMI.

PMI has been included in the agile approach (Lines: 241-243).

Point 12: The research method:

4 companies with classic methods and 4 with agile ones. 32 clients matching and 32 not matching....but which ones come from classic or agile? 16 per each?

Response 12: This issue has been further clarified in the subsection: The logic of research - survey scheme and tools (Lines: 330-340). Finaly, both groups are represented by 32 clients (of which 16 are serviced by the classic methodology and 16 - by the agile methodology).

Point 13: The research method:

More applications of Servperv should be included in order to consider this method.

Response 13: Indeed, it is worth pointing out why this method was considered and presenting also examples of its applications in various fields. The Servperv method is successfully used in contemporary research as a basis for constructing own measuring instruments in various areas. Examples of contemporary Servperv applications in various areas have been added in the manuscript (Lines: 286-288). The premises concerning the rightness of using this method in this study were also indicated (Lines: 278-288).

Point 14: The research method:

These A and B groups and divided according to their match with IT PM methodologies but, other characteristics from their companies (size, level of technology applied, investments, etc.) should be stressed in order to chech if the two sample of 32 clients are ok.

Response 14: It has been explained, that in this paper the client is understood to be the person selected for direct contact with the IT team. These were employees who, in the opinion of senior management, knew the areas of the organization that were relevant to the development of dedicated software. (Lines: 184-190) Then, each clients were selected so as to represent projects that were similar throughout the study in terms of the set criteria. Thus, the criteria characterizing the IT project were important. Their description has been added (Lines: 310-327).

Point 15: Research results:

A simple average study...

Response 15: Thanks for your claim.
The verification of the research hypothesis was based on comparing the differences between two independent groups. To illustrate the differences between them, a comparison of averages in the  dimensions of Servperv instrument was used. Indeed, detailed study results were not included with the manuscript. They have been included as supplementary material (Lines: 400-401).

Point 16: Discussion:

Sustainable approach is absolutely missing. 

Response 16: Indeed, I agree with you that sustainable approach should be included in the discussion.

It was included in the revised Discussion and Conclusions section (Lines: 437-446, 474-477) and additionally was indicated in the new subsections: Implications for practice (Lines: 459-465); Limitations and future research (Lines: 479-496).

Reviewer 4 Report

The real, main goal of this article is to investigate whether matching the choice of project management methodology to the level (type) of client increases customer satisfaction with the use of IT in the organization. Unfortunately, this goal is only correctly specified on lines 74-76. Earlier it is worded very vaguely in Abstract. So what is the actual purpose of the article? There is no proper rationale for what this problem has to do with sustainability in design. It is therefore doubtful whether the content of the article corresponds to its title.

Strengths

  • the article concerns of an interesting and always up-to-date subject of the dependence of customer satisfaction on the relationship between the customer type and the method of IT project management,
  • an attempt to apply metodyservpery in the economic reality.

Weakness and suggestions for improvement

  • There is no justification why this problem is presented in terms of sustainability.
  • The Summary lacks a clearly defined purpose, subject and purpose of research into results and conclusions.
  • In the Introduction, there is no precise presentation of the research problem, the author focuses on IT management problems.
  • According to many years of research by the Gartner Group, user problems have always dominated the problems of IT project management, the author does not seem to know it (despite being referred to the Gartner Report from 2018),
  • The introduction lacks an exact definition of the research gap and the description of the article structure,
  • The structure of the article is incorrect - there is no literature review and conclusions.
  • There is no precise definition of the client's category - this is a very broad category, even for SMEs.
  • There is a lot of outdated literature, a lot has changed since the beginning of the century.
  • The presented approach to the problem is only possible for the customer market.
  • It is not clear what the criteria for dividing the client into groups A and B are, it is rather arbitrary (no description of the semi-structured interview), Wysocki's work is from 2009.
  • The method from 2000 is used, an extension of the method from 1985. These methods were developed to evaluate classical design methods, not agile methods. Why does the author use such old methods and not even try to invent her own?
  • The juxtaposition of two commercial classic methods (PRINCE2 and PMBok with precisely formulated life cycle and procedures and 7 (!) Agile methods (sometimes even without a specific life cycle, except scrum) is not appropriate (wrong assumption).
  • Gartner Group reports say that agile methods are better for SMEs than classical methods. So why does the author research only small and middle enterprises?
  • Were the considered projects related to the Internet sphere? If so, the classical methods are not used there.
  • Why is it limited to new projects implemented from the beginning, and not implementation projects, which are the most numerous?
  • What are the rules for selecting people for direct contact?
  • How do you know that IT companies from the SME sector use either classical or agile methods?
  • What were the detailed criteria for selecting IT companies and their clients for analysis?
  • Why limited analysis to 8 clients from each company etc.?
  • Is it possible to determine which of them suits the client and which does not, before choosing the methodology?
  • What does a randomly selected IT team member know about the above subject?
  • How were the experts selected to evaluate the completion of the Survperv form?
  • What is the relationship between Survperv questions and the meto agile and classic life cycle (the life cycle is different in each agile method)?
  • For such a small sample, it is not necessary to calculate the Cronbach's alpha test.
  • Why was the Z test used to calculate the distance between solutions instead of e.g. the Euclidean distance?
  • Lack of conclusions, limitations specification and direction of further research.
  • The article should be checked by a native speaker.

All the above problems and doubts should be resolved.

Author Response

Response to Reviewer 4 Comments

Thank you very much for your effort in reviewing my manuscript. Thank you both for your positive words in the strengths part and for pointing out weaknesses and suggestions for improving them.

I have read all of your comments and claims very carefully and have tried to resolve all reported problems and doubts.

I have addressed all the issues you raised, as described below.
All changes to the manuscript are marked in green.

Point 1: The real, main goal of this article is to investigate whether matching the choice of project management methodology to the level (type) of client increases customer satisfaction with the use of IT in the organization. Unfortunately, this goal is only correctly specified on lines 74-76. Earlier it is worded very vaguely in Abstract. So what is the actual purpose of the article? There is no proper rationale for what this problem has to do with sustainability in design. It is therefore doubtful whether the content of the article corresponds to its title.

Response 1: I acknowledge that the purpose of the article has been worded very vaguely and without the context of sustainable approach. This weakness has been improved by more clearly formulating the purpose of the article and the goal of the study in Abstract (Lines: 12-27) and Introduction (Lines: 165-178) and addressing the sustainability context (Lines: 83-87, 120-167).

Point 2: There is no justification why this problem is presented in terms of sustainability.

Response 2: Indeed, the manuscript did not provide sufficient justification for considering the presented research problem in the context of sustainability. Therefore, the substantiation for this context has already been included in the abstract (Lines: 8-27) and the introduction indicates one of the two sustainability perspectives in IT projects in the context of which the research is presented (Lines: 83-87).

A literature review has been added which shows how the topic of sustainability in projects is discussed, with particular attention to the internal perspective of sustainability in IT projects (Lines: 120-159). It allowed to present a research gaps in this area (Lines: 160-173).

Point 3: The Summary lacks a clearly defined purpose, subject and purpose of research into results and conclusions.

Response 3: In order to clearly define purpose, subject and purpose of research as well as the results and conclusions, significant changes were made to the Abstract (Lines: 12-27).

Point 4: In the Introduction, there is no precise presentation of the research problem, the author focuses on IT management problems.

Response 4: The research problem was formulated in the summary of the research background. In the research background, a literature review has been added, which shows how the topic of sustainability in projects is discussed. In this way, the research gaps were identified and then the research problem was specified in the context of the identified gaps. Lines: 154-178

Point 5: According to many years of research by the Gartner Group, user problems have always dominated the problems of IT project management, the author does not seem to know it (despite being referred to the Gartner Report from 2018),

Response 5: Indeed, this has not been emphasized enough. User problems have indeed always dominated the problems of IT project management, but perspectives and approaches to overcome them have evolved. This has been included in the manuscript (Lines: 66-73).

Point 6: The introduction lacks an exact definition of the research gap and the description of the article structure,

Response 6: I admit that this has been omitted. In the research background a literature review  was added and on its basis the research gaps were defined (Lines: 160-173). The structure of the article is presented in the summary of the Introduction (Lines: 88-92).

Point 7: The structure of the article is incorrect - there is no literature review and conclusions.

Response 7: In the research background section, a new literature review has been added (Lines: 120-159) and the section title has been changed to: The research background and literature review. The conclusions from the research have been specified in the section Discussion and Conclusions (Lines: 434-446) and a new subsections have been added as part of the conclusion: Implications for practice (Lines: 458-477).

Point 8: There is no precise definition of the client's category - this is a very broad category, even for SMEs.

Response 8: Indeed, there was a lack of precision in defining the client category.

It has been explained, that in this paper the client is understood to be the person selected for direct contact with the IT team. These were employees who, in the opinion of senior management, knew the areas of the organization that were relevant to the development of dedicated software. (Lines: 184-190) Then, each clients were selected so as to represent projects that were similar throughout the study in terms of the set criteria (Lines: 310-327).

Point 9: There is a lot of outdated literature, a lot has changed since the beginning of the century.

Response 9: The literature has been updated by new items especially in a literature review (Line: 120-159) and in research methods (238, 243), and especially in examples of the contemporary use of the Servperv method in various industries (286-288).

Point 10: The presented approach to the problem is only possible for the customer market.

Response 10: The client plays a special role in IT projects. And in IT projects where the object is to develop dedicated software for organizations (as in this study), the client is extremely important to complete the project with full success. The significance of the proposed approach is raised in the subject literature in the context of sustainable development (Lines: 120-159).  The importance of the client's role, assessment of his maturity and the choosing of an adequate methodology of IT project management are included in the manuscript too (Lines: 191-192).

Point 11: It is not clear what the criteria for dividing the client into groups A and B are, it is rather arbitrary (no description of the semi-structured interview), Wysocki's work is from 2009.

Response 11: The research methods section has been significantly enriched with an explanation of the method of dividing clients into groups A and B.

The instruments used in the study: semi-structured interview and client maturity matrix were presented. The structure of the client maturity matrix is described (Lines: 191-199, 221-231) and presented in a new figure (Lines: 218-221). The using of a semi-structured interview is described (Lines: 204-216), and the interview itself is included in the supplementary materials (Line: 206-207). It was explained how the client maturity matrix can be related to methods (classic, agile), referring to previous research (Lines: 244-250). Finally, it was summarized how the clients were divided into two groups, A and B (Lines: 251-255).

Wyskocki in his work referred to the classic division of methodologies, but it is of course better to point to newer sources. More recent method classification references are included (Lines: 236).

Point 12: The method from 2000 is used, an extension of the method from 1985. These methods were developed to evaluate classical design methods, not agile methods. Why does the author use such old methods and not even try to invent her own?

Response 12: The Servperv method is successfully used in contemporary research as a basis for constructing own measuring instruments in various areas. Examples of contemporary Servperv applications in various areas have been added in the manuscript (Lines: 286-288). The proposed measuring instrument is a proprietary instrument dedicated to researching IT projects where the object is to development develop software for organizations (Lines: 362). The criteria used therein relate to areas occurring in IT projects in the software development category. In these areas, the level of client satisfaction with an IT project managed by a specific method is examined, and the different life cycles of methods alone are not compared (Lines: 362-366). The instrument was developed in consultation with experts in the field of IT project management methods in both categories - classic and agile (Lines: 353-361).

Point 13: The juxtaposition of two commercial classic methods (PRINCE2 and PMBok with precisely formulated life cycle and procedures and 7 (!) Agile methods (sometimes even without a specific life cycle, except scrum) is not appropriate (wrong assumption).

Response 13: As mentioned above, the Servperv measuring instrument has been developed to examine the level of client satisfaction with IT projects managed by a specific method in the software development project. This instrument does not juxtapose such different life-cycle methods. Collaboration with experts experienced in both method categories (classic and agile) has resulted in the development of a tool that can be used regardless of the method's life cycle, as noted in the manuscript (Lines: 362-366).

Point 14: Gartner Group reports say that agile methods are better for SMEs than classical methods. So why does the author research only small and middle enterprises?

Response 14: Indeed, such are the reports of the Gartner Group. Nevertheless, in practice, both agile and classic methods are used in Polish IT companies from the SME sector. Sometimes entire IT teams are trained in the PRINCE2 methodology for some projects. And as the study proves, after a deeper analysis of the issue, whether the agile method is better in all cases is not so clear-cut. It was indicated in the manuscript that:

  • Polish companies implementing IT projects (development dedicated software for organizations) are mostly companies from the SME sector (Lines: 307-309),
  • IT teams of these companies use a specific IT project management methodology, which can be classified as classic or agile methods (Lines: 303-304),
  • also, most of the companies that prefer custom software are from the SME sector (Lines: 328-329).

Point 15: Were the considered projects related to the Internet sphere? If so, the classical methods are not used there.

Response 15: The considered projects didn’t relate to the Internet sphere. The object of the considered IT projects was the development of software dedicated to the organization. Details of the  considered projects are provided in the manuscript (Lines: 310-327).

Point 16: Why is it limited to new projects implemented from the beginning, and not implementation projects, which are the most numerous?

Response 16: Interaction with the client in the IT project was essential for the assumed research objective. It is most clearly visible in IT projects, where the object is to develop dedicated software for organizations. Therefore, this category of IT projects was selected for the study, as outlined in the manuscript (Lines: 325-327). Moreover, these projects are numerous among Polish SMEs, which constitute the largest group of organizations in Poland.

Point 17: What are the rules for selecting people for direct contact?

Response 17: Organizations ordering the software themselves selected people for direct contact. These were employees who, in the opinion of senior management, knew the areas of the organization that were relevant to the development of dedicated software. Lines: 185-190

Point 18: How do you know that IT companies from the SME sector use either classical or agile methods?

Response 18: An important assumption of the study was the definition - naming by the IT company of the used method of IT project management. In this way IT company, depending on the methodology of project management indicated by each other was classified as a user of classic or agile methods. This is explained in the manuscript in the following lines: 301, 303-306.

Point 19: What were the detailed criteria for selecting IT companies and their clients for analysis?

Response 19: Indeed, these criteria were omitted in the manuscript. The detailed criteria and method of selecting IT companies and their clients for the study have been specified. This have been added. The criteria and method of selecting IT companies for the study are described in the manuscript in lines: 296-302. The criteria and method of selecting clients for the study were determined by the criteria for selecting IT projects, described in lines: 310-329.

Point 20: Why limited analysis to 8 clients from each company etc.?

Response 20: The number of clients from each company was limited by the number of projects that met the criteria assumed in the study (Lines: 310-327, 331-336). These were IT projects where the object is to develop dedicated software for organizations. Therefore, the research was also limited by the consent of these organizations to conduct the research. In addition, it was important that clients impressions of the IT project were fresh in their memory, so the projects could not be too old. All this influenced a limitation of surveyed clients.

Point 21: Is it possible to determine which of them suits the client and which does not, before choosing the methodology?

Response 21: The semi-structured interview and the client's maturity matrix can be successfully used to determine which IT project management methodology suite the client and which does not, before choosing the methodology. This was indicated in the manuscript (Lines: 256-259).

Point 22: What does a randomly selected IT team member know about the above subject?

Response 22: IT team members were not asked about this subject. The clients were examined using a semi-structured interview and the client's maturity matrix. This is outlined in the manuscript in lines: 251-255.

Point 23: How were the experts selected to evaluate the completion of the Survperv form?

Response 23: Indeed, the criteria and method of selecting experts for consultation in the development of the Servperv instrument were omitted. This was supplemented in the manuscript in lines: 353-361.

Point 24: What is the relationship between Survperv questions and the meto agile and classic life cycle (the life cycle is different in each agile method)?

Response 24: The Servperv measuring instrument examines the level of client satisfaction with IT projects managed by a specific method. The statements contained in it refer to areas occurring in IT projects in the software development category. Collaboration with experts experienced in both method categories (classic and agile) has resulted in the development of a tool that can be used regardless of the method's life cycle, as noted in the manuscript (Lines: 362-366).

Point 25: For such a small sample, it is not necessary to calculate the Cronbach's alpha test.

Response 25: The motivation to use Cronbach's alpha was to check the internal consistency of developed Servperv instrument. It is indeed important in this regard to pay attention to the sample size. One of the common issues in studies is to determining a sufficient sample size so that research conducted can consistency or stability of an instrument. The literature on the subject indicates that the sample size is at least 30 and this condition is met in this analysis. This thread and references where this subject is widely discussed, were added in the manuscript. Lines: 378-384

Point 26: Why was the Z test used to calculate the distance between solutions instead of e.g. the Euclidean distance?

Response 26: A Z-test (or Mann-Whitney U test) was conducted, which is used to compare differences between two independent groups when the dependent variable is either ordinal or continuous, but not normally distributed. We have a Likert scale for ordinal variables and about 30 observations in each subsample, so this nonparametric approach was used. The suggested use of the Euclidean distance is very interesting, but there is concern that a much larger size of the analyzed sample would be necessary.

Point 27: Lack of conclusions, limitations specification and direction of further research.

Response 27: These sections were indeed missing in the manuscript.

A Conclusion section has been added to the Discussion section (433-446) and new subsections have been introduced there:

Implications for practice (Lines: 458-477)

Limitations and future research (Lines: 478-496).

Point 28: The article should be checked by a native speaker.

Response 28: An order for English language editing of the manuscript by MDPI will be done.

Round 2

Reviewer 2 Report

Congratulations for the improvements on your work.

The paper, in its current version, would greatly benefit by the including a graphical abstract in the introduction. This would help the potential reader follow the logical discurse.

Furthermore, minor english spelling check is required.

After clearing these minor issues, the paper can be published.

Thank you and good luck!

Author Response

Response to Reviewer 2 Comments

I am very grateful for the time and energy you dedicate to review my paper. Thank you both for your positive words and suggestion to include a graphic abstract in the introduction.

Point 1: The paper, in its current version, would greatly benefit by the including a graphical abstract in the introduction. This would help the potential reader follow the logical discurse.

Response 1: I agree that a graphical abstract depicting the manuscript content logic can greatly improve the perception of the paper.
A graphic abstract has been added in the introduction in line: 91.

Point 2: Furthermore, minor english spelling check is required.

Response 2: An order for English language editing of the manuscript by MDPI will be done.

Reviewer 3 Report

The abstract is too long. Is correct the end of line 11? Please review it and try to synthesize it.

In the introduction, success and failure in IT projects are reviewed. However, the CHAOS reports from the  Standish group are missing. These reports are very valuable for the international community.

Cultural, traditional and legal special conditions from Polish context remain not stressed.

In traditional project management, the two most important associations are PMI and IPMA. PRINCE2 by axelos is only relevant in specific sectors and nations. In fact, ISO 21500 takes processes from PMI and competences from IPMA.

If the Servperv questionnaire is only used to measure the quality dimensions, this does not cover the most aspects of IT PM (line 349). Please, read literature contrasting ISO 10006  vs ISO 21500.

The process to make and check the questionnaire should be included into the paper. Why 5 areas? Why 48 criteria? Which requirements to be considered an expert? Why these competencies? This part is the most important one. However this is not highlighted properly. These questions must be solved in order to validate the questionnaire.

The Cronbach's alpha is the most widely used method for estimating internal consistency reliability. However, there are other ways to measure the reliability and goodness of likert scales...that authors must compare.

Author Response

Response to Reviewer 3 Comments

I am very grateful for the time and energy you dedicate to review my paper, considering weak points and suggestions for improving them. I have read all of your comments and claims very carefully and have tried to resolve all reported problems and doubts.

Below you can see the corrections I have introduced into my paper correspondingly.

New changes to the manuscript are marked in light blue.

Point 1: The abstract is too long. Is correct the end of line 11? Please review it and try to synthesize it.

Response 1: Indeed, the abstract was too long. The abstract has been shortened (Lines: 8-24). As a result of the synthesis, the previous content of line 11 was redrafted (Line: 10).

Point 2: In the introduction, success and failure in IT projects are reviewed. However, the CHAOS reports from the  Standish group are missing. These reports are very valuable for the international community.

Response 2: In the introduction, the CHAOS report from the Standish Group was referred to in the first paragraph (Line: 32 reference No. 18). The reports of this report concerning IT projects, referring to the subject of the undertaken research, were emphasized (Lines: 52-56, 62).

Point 3: Cultural, traditional and legal special conditions from Polish context remain not stressed.

Response 3: Indeed, the cultural, traditional and legal aspects from Polish context are not included. Thank you for pointing them out, as these conditions may also be important for understanding country-specific project management problems. This raised aspects have been added (Lines: 179-214).

At the beginning, the attitude of Polish IT companies to the application of approved project management standards was presented (Lines: 179-185). Then, the cultural and traditional aspects of IT companies in Poland were referred to, and the Polish educational system in the IT area was also mentioned (Lines: 186-193).

Regarding the legal context, there are no regulations typical of IT project management in Poland. However, there are legal aspects that may constitute a barrier or support for the IT sector. These legal conditions have been taken into account in lines: 196-204. The issues of supporting Polish SME IT companies in the field of digital development by the government were also raised (Lines: 205-209). This is an important aspect that may condition companies' investments in IT projects (Lines: 210-212).

Point 4: In traditional project management, the two most important associations are PMI and IPMA. PRINCE2 by axelos is only relevant in specific sectors and nations. In fact, ISO 21500 takes processes from PMI and competences from IPMA.

Response 4: Indeed, in the part about project management methodologies, it is worth point at the contribution of PMI and IPMA associations to shaping them and the ISO 21500 standard indicating the model of project integration with the organization's environment. Information about this have been added in the lines: 274-281.

Point 5: If the Servperv questionnaire is only used to measure the quality dimensions, this does not cover the most aspects of IT PM (line 349). Please, read literature contrasting ISO 10006  vs ISO 21500.

Response 5: I admit that the statement in line 349 of the previous version of the manuscript could be misleading. This has been changed to clarify that those are a Servperv quality dimensions, covering theorems related to clients' perception of IT project management (Lines: 391-392).

Indeed, if the study referred strictly to the normative dimensions of quality, it would be necessary to adopt the ISO standards.

There are many references in the literature on the applicability of ISO standards in the area of project management within an organization. ISO 10006 provides guidance on the application of quality management in projects, but it does not serve as a guide in project management itself. ISO 21500 standard goes further. This international standard indicates to the model of using the opportunities of an integrated strategy connecting projects and the organization's environment and translating them into benefits for the organization. However, the benefits are not obtained directly from the project implementation, but as a result of the organization's functioning after its implementation. In this perspective, the project is seen as a tool for improving the efficiency of the organization rather than as a working method. The model is very useful for companies reorganizing their manufacturing processes or implementing management concepts. It should be considered how it can be related to organizations for which project management is the basic form of activity (e.g. IT companies, consulting companies, development companies, design offices) and implement projects in the external environment. According to Professor Wawak, who has been studying both trends for many years: quality of management and project management, IT companies wanting to implement ISO standards along with the implementation of IT projects in external organizations may face many barriers. Professor also claims that the model proposed in ISO 21500 may also not be appropriate for some software development companies or applying management by projects.

The study presented in the manuscript concerns software development IT projects, implemented in external organizations by IT companies of a project nature. Therefore, it was decided within collaboration with an expert panel to develop an instrument to measure client perception and satisfaction in the area of IT project management, where the object is to develop dedicated software for organizations.

Point 6: The process to make and check the questionnaire should be included into the paper. Why 5 areas? Why 48 criteria? Which requirements to be considered an expert? Why these competencies? This part is the most important one. However this is not highlighted properly. These questions must be solved in order to validate the questionnaire.

Response 6: Indeed, the process of making and validating the questionnaire has not been included into the paper. Thank you for pointing out this shortcoming. This has been supplemented as follows.

The reasons for selecting experts with such competences and in this number were indicated (Lines: 395-396).

The criteria for experts for this study were presented in the following lines: 397-410.

Criteria were adopted that meet the requirements for the rank of expert, indicated in the literature on the subject (e.g. Dreyfus model). So an expert is a person who presents situational knowledge, a holistic perspective, intuitive decision-making processes based on his knowledge and experience, and a high level of commitment to the domain area. This is also confirmed by the IPMA Competency Development Model. The aspects that were important for the specificity of the study were listed, such as knowledge of project management methodologies from both groups (classic and agile) along with the confirmation criterion (certificates); experience in IT project management, where the object is to develop dedicated software for organizations; direct contact with client.

Then, the way of constructing of the Servperv measuring instrument within an expert panel was presented. The process of establishing a consensus, the scale of assessments and the principles of evaluation were indicated, referring to the literature on the subject (Lines: 411-456).

Point 7: The Cronbach's alpha is the most widely used method for estimating internal consistency reliability. However, there are other ways to measure the reliability and goodness of likert scales...that authors must compare.

Response 7: I agree with this statement that there are other ways to measure the reliability and goodness-of-fit (GOF) likert scales, such as for example Composite reliability (CR) or AVE. I know that there are many different the goodness-of-fit (GOF) to determine the accuracy of likert scale that are based on different distributions: Cramer-Rao lower bound test, likelihood ratio test, Wald statistic, and Langrange multiplier test. In the presented study, I limit myself only to comparing the two groups - 32 observations per group (Line: 325). In the next stage of my research, I plan to extend the research sample considerably and apply the Confirmatory Factor Analysis (CFA), along with again Cronbach Alpha, and obligatory with CR and AVE ratios. And also GOF test.

Reviewer 4 Report

I have no more critical comments. All my previous doubts and questions have been satisfactorily corrected.

Author Response

Response to Reviewer 4 Comments

I am very grateful for the time and energy you dedicate to review my paper. Thank you for your positive words.

Your previous comments and questions gave me the opportunity to significantly improve my manuscript.

Thank you very much.

Round 3

Reviewer 3 Report

The paper has been improved. Although there are some things I do not agree, author has justified it.